# Cost-effectiveness of Vitamin A supplementation among children in three sub-Saharan African countries: An individual-based simulation model using estimates from Global Burden of Disease 2019

**Aditya Kannan** *, **Derrick Tsoi, Yongquan Xie, Cody Horst, James Collins, Abraham Flaxman**

Institute for Health Metrics and Evaluation, University of Washington, Seattle, Washington, United States of America

* adityakannan@cmu.edu

**Data Availability Statement:** All relevant data are within the paper and its Supporting Information

## Abstract

### Background

Vitamin A Supplementation (VAS) is a cost-effective intervention to decrease mortality associated with measles and diarrheal diseases among children aged 6–59 months in low-income countries. Recently, experts have suggested that other interventions like large-scale food fortification and increasing the coverage of measles vaccination might provide greater impact than VAS. In this study, we conducted a cost-effectiveness analysis of a VAS scale-up in three sub-Saharan African countries.

### Methods

We developed an individual-based microsimulation using the Vivarium simulation frame-work to estimate the cost and effect of scaling up VAS from 2019 to 2023 in Nigeria, Kenya, and Burkina Faso, three countries with different levels of baseline coverage. We calibrated the model with disease and risk factor estimates from the Global Burden of Disease 2019 (GBD 2019). We obtained baseline coverage, intervention effects, and costs from a systematic review. After the model was validated against GBD inputs, we modeled an alternative scenario where we scaled-up VAS coverage from 2019 to a level that halved the exposure to lack of VAS in 2023. Based on the simulation outputs for DALYs averted and intervention cost, we determined estimates for the incremental cost-effectiveness ratio (ICER) in USD/ DALY.

### Findings

Our estimates for ICER are as follows: $860/DALY [95% UI; 320, 3530] in Nigeria, $550/DALY [240, 2230] in Kenya, and $220/DALY [80, 2470] in Burkina Faso. Examining the data for DALYs averted for the three countries over the time span, we found that the scale-up led

files. The data were generated from simulation. The code is public and can be reproduced from GitHub (https://github.com/adityak77/vivarium_conic_vitamin_a_supp).

**Funding:** ADF received funding from the Bill and Melinda Gates Foundation (https://www.gatesfoundation.org) under grant OPP-1170133. The funders had no role in study design, data collection and analysis, decision to publish, or preparation of the manuscript.

**Competing interests:** I have read the journal's policy and the authors of this manuscript have the following competing interests: ADF has consulted recently for Janssen; SwissRe; Sanofi; Merck for Mothers; and Agathos, Ltd. Other authors have no conflicts of interest to disclose.

to 21 [5, 56] DALYs averted per 100,000 person-years in Nigeria, 21 [5, 47] DALYs averted per 100,000 person-years in Kenya, and 14 [0, 37] DALYs averted per 100,000 person-years in Burkina Faso.

## Conclusions

VAS may no longer be as cost-effective in low-income regions as it has been previously. Updated estimates in GBD 2019 for the effect of Vitamin A Deficiency on causes of death are an additional driver of this lower estimate of cost-effectiveness.

## Introduction

Vitamin A deficiency (VAD) is a risk factor for several major causes of death in children, including diarrheal diseases and measles [1]. Vitamin A supplementation (VAS) has historically been an intervention of interest for the prevention of xerophthalmia and night blindness as well as reduction of child mortality caused by measles and diarrhea in many low-income countries [2, 3]. Globally among children under 5, VAD is responsible for 9.9% [95% UI; 8.3, 11.6] of deaths due to diarrheal diseases and 12.0% [0.5, 23.1] of deaths due to measles [1]. Over the past few decades, efforts to increase coverage have decreased the burden of disease in children [1, 3]. From 1995 to 2005, the number of preschool-age children with VAD dropped from 251 million to 190 million globally [3]. However, in 2019, over 3 million disability-adjusted life years (DALYs) were still lost due to VAD globally [1]. Reducing the prevalence of VAD in children under 5 years of age could be an important part of the strategy to meet Sustainable Development Goal 3.2 of reducing child mortality to at most 25 per 1,000 live births by 2030 [4].

VAS has been an intervention that was popular for its low cost, varying between 0.50–1.50 USD per capsule based on spatial and temporal differences [5–7] while maintaining high efficacy in lowering VAD. It is commonly distributed as part of Child Health Days (CHD), which is a mass campaign designed to deliver immunizations and nutritional supplements to children. CHDs often last for a week, take place semiannually, and are a national effort to locate children though community-based or outreach campaigns. Two supplements every year for children up till the age of five is enough to lower the risk of death associated with deficiency [8]. Systematic reviews have shown that the relative risk of VAS on all-cause mortality and on night blindness are 0.88 [0.83, 0.93] and 0.32 [0.21, 0.50], respectively [9].

In recent years, some experts have suggested providing other interventions like the measles vaccine and large-scale food fortification in conjunction with VAS in order to reduce the incidence associated with measles and diarrhea [9]. Experts still vigorously debate the best course of action to tackle VAD moving forward [10]. There is very little recent literature on the cost-effectiveness ratio of VAS, and many of these papers do not consider VAS individually but rather as one among a group of interventions [11]. Using simulation results, we can determine the cost-effectiveness of VAS as a standalone intervention. Leveraging data from the Global Burden of Disease (GBD) 2019 study, we set out to estimate the cost-effectiveness of VAS in an individual-based model using the Vivarium microsimulation framework.

With a Vivarium model calibrated to estimates from the GBD 2019 study, we projected the impact of increasing coverage of VAS in three sub-Saharan African countries with different levels of existing coverages: Burkina Faso, Kenya, and Nigeria. We aimed to estimate the impact of scaling up this intervention by measuring the change in exposure to VAD and the

corresponding change in DALYs from year to year. We compared the outcomes of the interventions in the three countries to investigate how different levels of baseline coverage affect the cost-effectiveness of scale-ups. Furthermore, we calculated the DALYs averted and costs of the scale-up to add up-to-date, quantitative evidence to support decisions on which interventions would be worth pursuing.

## Materials and methods

### Vivarium model

We used Vivarium, an individual-based, discrete-time, Monte Carlo simulation framework developed by the Simulation Science team at the Institute of Health Metrics and Evaluation [12]. In the Vivarium model, we track a population of simulants over the period of the simulation. For each individual simulant, we track whether they are supplemented, have been exposed to VAD, have a disease, and are alive on every day of the simulation. With this framework, we can calculate the cost-effectiveness of increasing the coverage of VAS by running two distinct simulations. First, we run a baseline simulation that reflects the scenario where the coverage of VAS remains constant over the duration of the simulation. Next, we simulate the alternative scenario where the coverage of VAS increases steadily over the course of the simulation. We track the DALYs and cost of each scenario, and we can then obtain the DALYs averted and additional cost.

Vivarium includes modular components that incorporate distributions and data from the Global Burden of Disease (GBD) [1]. These components include an intervention model, a risk exposure model, a risk-effect model, a cause model, a mortality/morbidity model, and a cost model (**Fig 1**). Each individual component requires inputs to be modelled. For example, the risk exposure model would need risk exposure prevalence; the risk-effect model would need relative risks for risk-cause pairs; the cause model would need disease prevalence and incidence; the mortality model would need cause-specific mortality rates and all-cause mortality rates. We used incidence, prevalence, mortality, and relative risk data from GBD 2019 to calibrate the risk-effect model, the cause model, and the mortality/morbidity model. We conducted a meta-analysis to determine the inputs in the other models.

VAS in children is meant to reduce VAD, which in turn, affects measles and diarrheal diseases (**Fig 1**). We used Vivarium to model the two causes of death (measles and diarrheal diseases) as well as the risk factor of Vitamin A deficiency by calibrating the simulation to estimates from GBD 2019 that produced relative risks for risk-cause and cause-outcome pairs. We use compartmental models to describe the dynamics of causes of death. The measles component utilized a Susceptible-Infected-Recovered (SIR) model with a 10-day duration for the

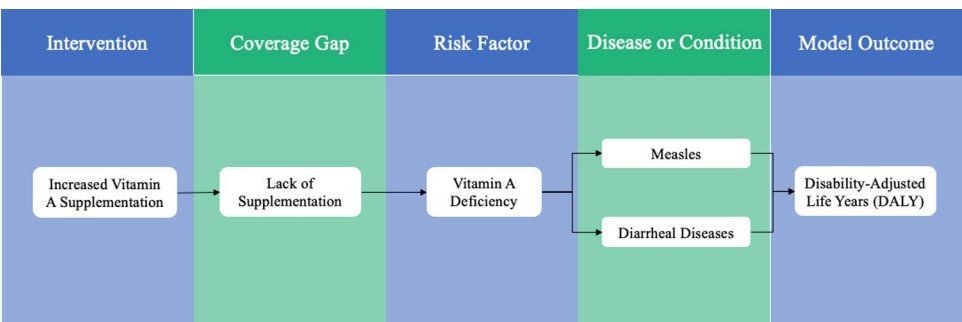

**Fig 1. Microsimulation concept model.**

infected state. While all individuals are initially susceptible to measles, some fraction may be exposed to it over time. Those who recover from measles will never be susceptible to it again, so we use an SIR model to describe it. The diarrheal disease component used a Susceptible-Infected-Susceptible (SIS) model because individuals who get diarrhea will still be susceptible to it after they recover. At each time step, based on the appropriate health models and country-level data from the GBD database, the simulation determined whether the state of each individual changed with regards to either of the two diseases. We chose a step size of 1 day to be able to capture the possibility of being inflicted with a disease on successive days in the case of diarrhea.

At model initialization, Vivarium assigns the age, sex, and VAD status of simulants. VAD is determined as a serum retinol concentration of less than 0.70 μmol/L. We model VAD as a binary categorical variable. At each time step, the simulation samples probability distributions using a Monte Carlo method to determine changes in a simulant's status. These changes include whether an individual receives VAS, experiences incidence of diarrheal disease, measles, or changes to their VAD status.

First, we determined the initial characteristics of the population that we would simulate. We used an open cohort of 1,000,000 individuals between the ages of 0 and 59 months. While we tracked individuals in the simulation starting from birth, we only simulated the effect of supplementing individuals between 6–59 months, the ages eligible for VAS. We stopped tracking simulants whose age exceed 59 months. For both the baseline and alternative scenarios, we ran one simulation each from 2017 until 2023. For the alternative scenario, we modeled a scale-up of VAS coverage starting in 2019. We allowed the alternative simulation to start two years prior to scale up to verify that the model outputs were calibrated to the GBD 2019 estimates relevant to Vitamin A. When determining the impact and cost-effectiveness of the intervention, we only consider the time period from 2019 to 2023 in the simulations.

Although the GBD 2019 Study includes estimates of the exposure and relative risk of VAD, as well as the coverage of VAS, it does not include information on the effectiveness of VAS to reduce VAD. We modelled VAS as a dichotomous attribute, with one possibility being the simulant received two doses of Vitamin A in one year and the other being that they did not receive the supplementation at all. We modeled the scale-up of coverage of VAS as stepping up from the baseline to the target value at equal intervals each year from 2019 to 2023 (held constant throughout each year). We determined the baseline coverage and intervention effectiveness using a systematic review. We employed a fixed-effect meta-analysis to synthesize the values in the literature review (see Vitamin A Supplementation Coverage and Vitamin A Supplementation Effect sections below). A fixed-effect meta-analysis assumes that our studies are measuring a shared effect value that deviates from study to study due to only sampling error. To generate values for coverage and relative risk, we weighted the values from different studies based on their sample sizes, where studies with more individuals have larger weights.

Finally, we calculated the cost for the intervention from the perspective of the group distributing the doses, including the costs of the capsule, training, and transportation. We determined the number of doses supplied by the simulation and multiplied that value by a unit cost. The unit cost was inferred from a literature search on VAS [5–7].

## Simulation inputs

**Vitamin A supplementation coverage.** For our estimate of VAS baseline coverage, we considered using the UNICEF database of VAS coverage by country for our estimates. However, due to the variation of estimates from year to year for a few of our countries of interest (specifically Nigeria and Kenya), we decided to use other sources.

GBD has a covariate for VAS coverage measuring the proportion of children who received one dose of Vitamin A over the last six months. In other work, this measure has been used as a proxy for the coverage of two doses over the past twelve months [8]. We used the GBD covariate and a separate literature search to determine the baseline coverage for each location. If the results of the literature search were similar to the GBD covariate, we used the value of the GBD covariate for the baseline coverage.

For the literature search, we included DHS surveys and articles from Google Scholar and PubMed. We included only sources that defined supplementation to be two doses within twelve months or one dose in six months. We used a fixed-effect meta-analysis based on each source's coverage and sample size to determine the value for the baseline coverage in each region [13]. For sources that did not provide a sample size, we assigned to them the smallest sample size of all the other papers.

The target coverage in all three countries was chosen so that the exposure to the lack of VAS was halved to represent an intervention scale-up of similar relative intensity for each country. We ran simulations for two scenarios: one representing the "business as usual" scenario with VAS coverage held constant at the 2017 level until 2023, and the other representing the VAS scale-up from the existing coverage in 2019 to the target coverage by 2023. We compared the deaths and DALYs in the scenarios in each country to find the impact of VAS scale-up.

**Vitamin A supplementation effect.** To determine the efficacy of VAS in reducing VAD as a risk factor, we performed a literature search of online articles. We used a combination of the search terms "Vitamin A supplementation," "Vitamin A deficiency," and "relative risk" in Google Scholar and PubMed to locate relevant articles. Of the articles found in these databases, we filtered out the ones that did not use the standard definitions for VAS (frequency of capsule distribution is twice per year) and VAD (serum retinol concentration less than 0.70 μmol/L) [8]. Additionally, we included articles that provided a value for the relative risk of VAS on VAD and excluded those that provided only relative risk of other outcomes like mortality, night blindness, and causes of death (more details in **S1 Appendix**).

We quantified the effect of a lack of VAS on VAD in terms of the relative risk, meaning the ratio of VAD prevalence among those without VAS to those with VAS. By conducting a fixed-effect meta-analysis on these online sources, we determined distributions for our simulation to decide which individuals would be afflicted with VAD based on their supplementation status.

**Intervention unit cost.** We calculated the overall cost of both the baseline and alternative scenarios by finding the number of doses distributed in the simulation and multiplying that by a unit cost. We created an observer in the simulation that reported the total number of supplemented years. One supplemented year corresponds to an individual who was alive, between 6–59 months old and was provided the two doses of Vitamin A over a one-year time period. We doubled the number of supplemented years to determine the number of doses distributed in a particular year of the simulation.

We determined the unit cost from a recent study on costs of Vitamin A interventions for the semiannual CHD program [7]. It considered costs for the capsule, training, wages, and transportation to provide the unit cost per dose. After adjusting this value by inflation, we arrived at a cost per dose of $0.60 USD. We took the product of the simulation's doses and the unit cost to determine the overall cost per scenario. The additional cost of the intervention was calculated by comparing the costs of the baseline and intervention scenarios.

## Analysis of simulation results

We tallied the population stratified on sex, year, and age with regards to our outcomes of interest, including years of life lost (YLL) and years lived with disability (YLD). From our simulation we added the number of YLLs and YLDs to find the total quantity of disability-adjusted life years (DALYs) throughout the simulation. We arrived at the number of DALYs averted in each location by finding the difference between the baseline and alternative scenarios. Similarly, we calculated the additional cost of the intervention in each region. Using these two values, we estimated the Incremental Cost-effectiveness Ratio (ICER), which is ratio of the change in cost to the DALYs averted. The ICER is useful for putting the health impact in perspective of the additional cost of the intervention.

The main two sources of uncertainty that we model in our simulation are parameter uncertainty and stochastic uncertainty.

We modelled our input parameters from GBD and literature searches as probability distributions. We utilized a Monte Carlo technique to draw samples from the distributions to generate random values. For each input, we propagated parameter uncertainty through the model and captured the outcomes (including DALYs averted, additional cost, and ICER) for each draw. We used a total of 100 draws for each country and took the median to represent the results for each scenario. We chose to summarize our results using the median instead of the mean as the median is resistant to outliers. Using the outcomes of the 100 draws, we also generate an uncertainty interval for the DALYs averted, additional cost, and ICER values.

Stochastic uncertainty is meant to model the variability across scenarios among identical individuals within the population that have the same age and sex. We wanted to run the baseline and alternative scenarios with the same stochastic variation, to allow us to examine the differences in the outcomes due to parameter uncertainty, while minimizing random noise. Vivarium uses common random numbers to reduce this variance between individuals [14]. Common random numbers ensure that the values drawn from an input parameter distribution are the same for both the baseline and alternative scenario in each draw.

## Verification and validation

We validated our simulation by comparing its estimates for health outcomes in the years 2017–2018 with the data collected in the GBD 2019 study. For each of the three countries, we compared the rate of YLLs and YLDs with regards to the two diseases that are affected by VAS. For each metric and disease, we stratified our results by sex and by age group (post neonatal, and 1–4 years) as we believed that these variables would substantially affect the rate of YLLs and YLDs.

In addition, we performed some more validations over the intervention years. We confirmed the coverage of VAS scales up in the alternative scenario and remains constant in the baseline scenario. We compared the relative risk of VAD with the value provided by GBD and verified that the effectiveness of VAS on reducing VAD matches the relative risk given by the meta-analysis.

## Results

### Simulation inputs

We found that the existing literature backed the GBD covariate for coverage in Burkina Faso [1, 15–18], so we used the GBD value for our simulation input. However, for Kenya and Nigeria, we judged the GBD covariate for supplementation coverage to be outdated and used the results of the literature search.

**Table 1. GBD 2019 estimates for risk factors and causes associated with VAD by country for ages 0–5.**

| | Prevalence (%) | | | PAF of VAD with respect to Cause | | DALYs by Cause (per 100K Person-Years | | |
|---|---|---|---|---|---|---|---|---|
| Nation | VAD | Measles | Diarrheal Diseases | Measles | Diarrheal Diseases | VAD | Measles | Diarrheal Diseases |
| Nigeria | 9 (5, 13) | 0.13 (0.05, 0.31) | 3.3 (2.7, 3.9) | 4.8 (0.2, 10.8) | 1.6 (0.2, 3.3) | 69 (43, 107) | 75 (1, 150) | 547 (83, 1,163) |
| Kenya | 48 (34, 61) | 0.05 (0.02, 0.12) | 3.0 (2.5, 3.6) | 17.2 (0.9, 32.6) | 5.2 (1.0, 9.7) | 170 (104, 257) | 86 (1, 253) | 757 (104, 1190) |
| Burkina Faso | 35 (25, 47) | 0.04 (0.01, 0.09) | 4.1 (3.3, 4.9) | 11.9 (0.5, 24.4) | 4.3 (0.7, 8.5) | 269 (160, 402) | 525 (9, 1581) | 877 (137, 1988) |

Abbreviations: GBD = Global Burden of Disease study; PAF = Population Attributable Fraction; VAD = Vitamin A Deficiency; DALY = Disability-Adjusted Life Year

Overall, we found that the baseline coverage for Nigeria was 32.1% [31.8, 32.3] [19–26], for Kenya it was 55.2% [54.7, 55.7] [25–31], and for Burkina Faso it was 88.4% [85.3, 91.9] [1]. The values we used in the simulation for baseline coverage were 32%, 55%, and 88% for Nigeria, Kenya, and Burkina Faso, respectively. As a result, for Nigeria the target coverage was 66%, for Kenya it was 77%, and for Burkina Faso it was 94%.

Our literature search for the relative risk of lack of VAS on the risk factor VAD provided 1,020 hits in Google Scholar and 33 hits in PubMed. After checking the articles based on our eligibility criteria, four sources eventually went into the fixed-effect meta-analysis. We found that the relative risk was 1.48 [1.05, 2.08] [32–35].

We used GBD 2019's estimates to model the relationship between risk factors and causes as well as the relationship between causes and outcomes (**Table 1**) [1].

## Simulation outputs

The simulation results show benefits of scaled-up interventions in each of the countries involved. Over the seven-year period, 21 [5, 56] DALYs per 100,000 person-years were averted in Nigeria, 21 [5, 47] DALYs per 100,000 person-years were averted in Kenya, and 14 [0, 37] DALYs per 100,000 person-years were averted in Burkina Faso (**Table 2**).

As coverage increased, the DALYs averted per year rose as well for all countries (see **S2 Appendix** for yearly simulation outputs). In 2017 and 2018, the number of DALYs averted were zero, when supplementation was yet to begin. Starting from 2019, the DALYs averted increased at a constant rate in all three regions depending on the size of the yearly coverage increase.

For each nation, the simulation records the number of supplemented years (the total number of person-years the simulation decided to supplement individuals with the intervention) which takes into account the current coverage. As the scale-up was smallest in Burkina Faso, the additional cost to implement the intervention was the least while Nigeria had the greatest cost. The additional cost per 100,000 person-years amounted to $17,880 [17,800, 17,960] in Nigeria, $11,390 [11,350, 11,430] in Kenya, and $3,100 [3,060, 3,120] in Burkina Faso

**Table 2. Overall outcomes for simulated intervention by country.**

| Country | Overall DALYs Averted Per 100,000 Person-Years | Overall Additional Cost Per 100,000 Person-Years (USD / 100,000 Person-Years) | Cumulative ICER (USD/ DALY) |
|---|---|---|---|
| Nigeria | 21 (5, 56) | 17,880 (17,800, 17,960) | 860 (320, 3530) |
| Kenya | 21 (5, 47) | 11,390 (11,350, 11,430) | 550 (240, 2230) |
| Burkina Faso | 14 (0, 37) | 3,100 (3,060, 3,120) | 220 (80, 2470) |

Abbreviations: DALY = Disability-Adjusted Life Year; USD = US Dollar; ICER = Incremental Cost-effectiveness Ratio

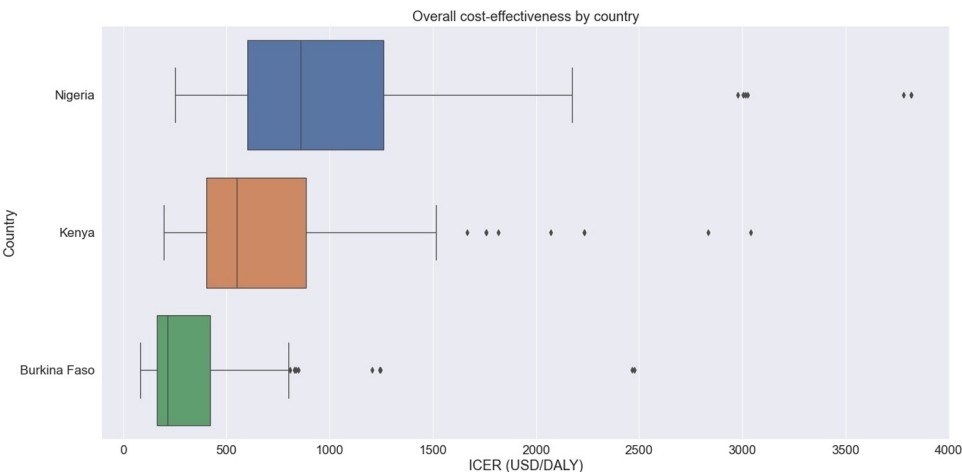

**Fig 2. Cost-effectiveness of intervention scale-up by country.**

(**Table 2**). The additional cost per year increased at a rate proportional to the increase in coverage for each specific intervention.

Despite averting the fewest DALYs over the course of the simulation, the intervention in Burkina Faso has the lowest median ICER among the countries. Overall, the intervention over the seven-year period would have an ICER of $860/DALY [320, 3530] in Nigeria, $550/DALY [240, 2230] in Kenya, and $220/DALY [80, 2470] in Burkina Faso (**Fig 2**).

## Validation

We validated the simulation output for each nation in 2017 and 2018 by comparing it to GBD estimates from that year. We stratified the simulation results by age, sex, and cause for the comparison. In particular, the two age groups that we validated were post neonatal and 1–4 years of age since the intervention is only provided to children between 6–59 months of age. The mean rate of YLLs and YLDs from that year over 100 draws are within 10% of the GBD value for both age groups and sexes for measles. Those values for diarrheal diseases are within 20% of the average in Nigeria and Burkina Faso but are well within the range of the parameter uncertainty estimated by GBD.

## Discussion

By comparing the effect of a scale-up of VAS coverage in three countries with varying baseline coverages, we measured the changes in the cost-effectiveness of different levels of scale-ups. In Nigeria, where the baseline coverage was the smallest, we simulated an intervention coverage scale-up from 32% to 66%, while in Burkina Faso, where the baseline coverage was the largest, we simulated an intervention coverage scale-up from 88% to 94%. The model found that scale-ups in all three nations produce similar rates of DALYs averted, all with overlapping uncertainty intervals. A surprising result is that although Nigeria had a larger scale-up than Kenya, the two nations had the same median DALYs averted. This can be attributed to the increased prevalence of VAD in Kenya compared to Nigeria (see **Table 1**).

In addition, the model predicted that Nigeria has the most additional cost whereas Burkina Faso has the fewest. Similarly, the intervention in Burkina Faso is the most cost-effective and the scale-up in Nigeria is the least. However, we note that the model's estimates for ICER have wide and overlapping uncertainty intervals. Because ICER is the ratio between additional cost

and DALYs averted, one factor alone may not sufficiently reflect the trend in the cost-effectiveness of an intervention. Although Burkina Faso had the fewest DALYs averted, the intervention had the lowest ICER because it has the smallest cost. There is substantial uncertainty in the DALYs averted and ICER values as well, but in all settings the median ICER is between 100–1000 USD/DALY.

Overall, our simulation study finds that VAS is not as cost-effective as previously reported in the literature. The cumulative ICER estimates for the nations in our simulation are approximately ten times larger than previous cost-effectiveness analyses of this intervention [11, 36]. Edejer's study from 2005 considers the cost-effectiveness of different combinations of nutrition-related interventions in sub-Saharan Africa. The authors found that VAS, Zinc Supplementation, Measles vaccination, and pneumonia case management together have an ICER of $85/DALY [11]. However, this study measures the effectiveness of VAS combined with other interventions, and therefore may not be directly comparable with our results. Another analysis by Chow and colleagues in 2010 found that VAS by itself had an ICER ranging between $23-50/DALY within India [36]. The differences in these values compared to our simulation's outputs may be due in part to the combination of interventions or differing locations of the studies.

However, we believe that the increase in ICER can be mainly attributed to the decrease in VAD over time and new estimates for the risk effect of VAD on causes of death. For example, among children under 5 years of age in Nigeria, VAD was responsible for 5,850 DALYs per 100,000 person-years in 2005 and 1,860 DALYs per 100,000 person-years in 2010, but it only caused 690 DALYs per 100,000 person-years in 2019 [1]. A lower prevalence of VAD would allow fewer DALYs to be averted, which would result in a larger ICER.

Changes in the modelling of VAS in GBD 2019 might have further reduced the impact of VAS. GBD 2019 used a new meta-analysis method for micronutrient modelling that was not applied in GBD 2017 [37]. The results of the new modelling approach reduced the effect of VAD on measles and diarrheal diseases by an order of magnitude [1, 38]. GBD 2019 also found that lower respiratory infections (LRI), a cause of death associated with VAD in GBD 2017, did not have sufficient evidence to estimate a causal relationship with VAD [1]. As a result, we chose not to include LRI in our model. We ran a separate simulation that was calibrated to GBD 2017 estimates and included LRI. This model found that ICERs were more similar to previous literature: Nigeria had an ICER of 41 [26, 61], Kenya had an ICER of 62 [46, 97], and Burkina Faso had an ICER of 35 [22, 59]. This illustrates the importance of the inputs to our model and how it can affect the results of our simulation.

Our cost-effectiveness analysis has two major strengths: the flexibility of the Vivarium framework, and the ability to incorporate uncertainty into the model. The Vivarium framework allows us to create models that are very flexible while calibrating population-level parameters to GBD estimates. The model used in our simulation only considered VAS, which is sometimes not possible in the field because individuals may be subject to multiple interventions simultaneously. In fact, much of the literature on cost-effectiveness of nutritional supplements combines different interventions when estimating the ICER. Additionally, the microsimulation model allows us to vary the number of draws and individuals in each simulation. This gives us the ability to incorporate uncertainty in our simulations.

Our approach has two limitations due to the scope of our model. In addition to lowering VAD by increasing VAS coverage, campaigns often boost the intake of VA as well. While GBD estimates for VAD include data on VA intake, we did not model VA intake explicitly. Second, our model does not incorporate the YLDs stemming from xerophthalmia and night blindness. However, the YLDs are small compared to the YLLs of the causes that we did include, so we believe that this omission does not greatly affect our results.

Another area for future improvement is our cost model. We determined the overall cost of the intervention using a singular value for the average unit cost of VAS (see **S1 Appendix** for more details on the literature search). We used sources from journal publications only, but there are other sources that claim the unit cost might be larger. The WHO One Health Tool estimates that the unit cost can range between \$2.45–3.17 USD. Increases in unit cost would create a proportional increase in ICER. For example, if the unit cost of VAS were five times larger, the additional cost and ICER would increase by a factor of five as well. Another drawback of our cost model is that it assumes that the intervention cost increases linearly as coverage expands. However, costs are likely to vary in different nations and even subnational regions based on accessibility, budget, and the size of the program. As a result, cost has a superlinear relationship with intervention coverage in the real world. Although our simulation can take uncertainty into account, it might not have enough complexity to consider all the factors of a realistic cost model. Up-to-date cost data as well as studies analyzing the nonlinear costs of supplementing individuals in isolated regions would improve the quality of this analysis.

## Conclusion

In this study, we calculated the ICER as a measure of the cost-effectiveness of scale-ups of VAS coverage in three countries using an individual-based microsimulation calibrated to match GBD estimates at the population level. We found that the intervention was not as cost-effective as it has been reported previously. This is due to falling levels of VAD among children in our countries of interest and lower risk effects for causes of death due to VAD in GBD 2019.

## Supporting information

**S1 Appendix. Simulation inputs.**
(DOCX)

**S2 Appendix. Yearly results.**
(DOCX)

## Acknowledgments

We would like to thank Dr. Nicole Young for review of the manuscript.

## Author Contributions

**Conceptualization:** Aditya Kannan, Derrick Tsoi, James Collins, Abraham Flaxman.

**Data curation:** Aditya Kannan, Derrick Tsoi.

**Formal analysis:** Aditya Kannan, Abraham Flaxman.

**Funding acquisition:** Abraham Flaxman.

**Investigation:** Aditya Kannan.

**Methodology:** Aditya Kannan, Derrick Tsoi, Abraham Flaxman.

**Software:** Cody Horst, James Collins.

**Supervision:** Abraham Flaxman.

**Validation:** Yongquan Xie.

**Visualization:** Aditya Kannan, Yongquan Xie.

**Writing – original draft:** Aditya Kannan.

**Writing – review & editing:** Aditya Kannan, Abraham Flaxman.

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
