## [Decision Letter · Decision Letter 0]

22 Dec 2021

PONE-D-21-34978Cost-effectiveness of Vitamin A Supplementation among children in three sub-Saharan African countries: an individual-based simulation model using estimates from Global Burden of Disease 2017PLOS ONE

Dear Dr. Kannan,

Thank you for submitting your manuscript to PLOS ONE. After careful consideration, we feel that it has merit but does not fully meet PLOS ONE’s publication criteria as it currently stands. Therefore, we invite you to submit a revised version of the manuscript that addresses the points raised during the review process. The reviewers have provided helpful advice as to how to rework the paper. One reviewer has strong expertise in the nutritional aspects, and a key point that is made is that the paper does not consider up-to-date evidence on mortality, and also that the paper uses outdated GBD estimates. The other reviewer has strong expertise in modelling economics of nutrition provides helpful suggestions on the modelling methodology and also the exposition to an audience (for PLOS) that is not steeped in econometric terms as used by economists, as well as the previous reviewer's point about using updated mortality rates from more recent studies. All the suggestions are important; the only one that I do not particularly agree with is changing the measurement of cost per DALY, to cost per death averted. Given the conventional thresholds, I believe that it is better to retain the analysis in cost per DALY averted.

We look forward to receiving your revised manuscript.

Kind regards,

Susan Horton

Academic Editor

PLOS ONE

Journal Requirements:

2.Thank you for stating the following in the Funding information of your manuscript: 

(The Bill and Melinda Gates Foundation (OPP-1170133) funded this project. The funders had no role in the decision to publish.)

(ADF received funding from the Bill and Melinda Gates Foundation (https://www.gatesfoundation.org) under grant OPP-1170133. The funders had no role in study design, data collection and analysis, decision to publish, or preparation of the manuscript.)

Reviewers' comments:

Reviewer's Responses to Questions

**Comments to the Author**

1. Is the manuscript technically sound, and do the data support the conclusions?

Reviewer #1: Partly

Reviewer #2: No

2. Has the statistical analysis been performed appropriately and rigorously? 

Reviewer #1: I Don't Know

Reviewer #2: No

3. Have the authors made all data underlying the findings in their manuscript fully available?

Reviewer #1: Yes

Reviewer #2: Yes

4. Is the manuscript presented in an intelligible fashion and written in standard English?

Reviewer #1: Yes

Reviewer #2: No

5. Review Comments to the Author

Reviewer #1: Please see the attached - as this version does not include the hyper links the reviewer provided:

The paper is very well thought out, very relevant, and timely in its topic. VAS is an essential child survival strategy in populations where VAD is a public health problem among children 6-59 mos of age. Despite it being very effective to reduce morbidity and mortality associated with VAD, it is an intervention that has been in place in some countries for over 20 years, and commitment to maintain such programs is waning in some countries. Cost-effectiveness data is critical to engaging policy makers, and advocating for continuation of this life-saving public health intervention especially as health systems are being stretched to the limits of their resources. For this reason we really welcome this analysis, and the rigour to which the authors brought to it.

Understanding that the authors simulated the scale up of VAS coverage from a hypothetical baseline of coverage to demonstrate that across three different settings, the effort to scale up the intervention is very cost effective.

I recommend however that this paper be revisited to consider the following:

1) The GBD 2019 significantly revised the methodology used and cited the following major changes that resulted in a very large reduction in # of DALYs due to VAD (reduced to 3.3M globally, and specifically 2.63M DALY’s in children U5):

• Vitamin A deficiency and vitamin A supplementation were modelled in ST-GPR to achieve improved time trends.

• Vitamin A supplementation estimates are now age-sex specific since supplementation campaigns target children.

• The age-specific stunting SEV was added as a covariate for vitamin A deficiency, alongside the three used last year: SDI, the availability of retinol activity equivalent (RAE) units in foods, and newly updated vitamin A supplementation.

• The evidence on vitamin A deficiency as a risk factor for diarrhoea, measles, and lower respiratory infections (LRIs) was re analysed and evaluated using MR-BRT. LRIs were removed as an outcome due to insufficient evidence, and the relative risks for diarrhoea and measles were updated. Notably, we no longer adjust relative risks for background vitamin A deficiency prevalence.

Attached to this review is the Lancet ref, as well please see this recently published discussion paper: Basis for changes in the global burden of disease estimates related to vitamin A and zinc deficiencies in the GBD 2017 and 2019 studies | Public Health Nutrition | Cambridge Core

Could the authors consider re-running the analysis with the GBD 2019 data? If not, I strongly recommend this be acknowledged in the introduction as well as in the Discussion as a limitation, and consider what the ICER would be if the GBD 2019 was the source instead.

2) Line 67 references a paper that compared measles vaccination and therapeutic VAS in reducing mortality due to measles. While I am unable to access the full paper referenced (#10), it appears that that specific paper was considering the number of capsules given to treat a case of measles (in accordance with WHO Guidance on the therapeutic uses of VAS) – whereas the authors are comparing this to the two doses provided in PREVENTIVE programs. While preventive VAS can reduce incidence of measles by 50% (Imad et al 2017) – it is not found to reduce mortality due to measles.

3) Line 67, in addition to the above also has the following error: Large scale Food fortification does not reduce deaths associated with measles, LRI and diarrhoea – instead it is a strategy to increase the consumption of Vitamin A through the daily diet, and as such, can be considered as a means to reduce VAD in the population over time. Vitamin A supplementation on the other hand, only temporarily reduces VAD in the population, and so if a child consumes two high dose supplements, the supplement is protecting them from the effects of VAD (caused by the low intake of VA from the diet). As a result, VAS and fortification (and other dietary strategies) are complementary to each other – not alternatives. The use and interpretation of serum retinol distributions in evaluating the public health impact of vitamin A programmes | Public Health Nutrition | Cambridge Core

4) Line 72 – refers to VAS as a treatment – however VAS programs are a preventive public health intervention. Recommend replacing “treatment” with the term “intervention”

5) Figure 1: this would change if GBD 2019 were considered instead

6) Line 128 – the cohort is referred to as children 0-59 months of age – however the eligible ages for VAS is 6 months to 59 months of age.

7) Line 139: VAS does not treat VAD. Recommend the change to “effectiveness of VAS to reduce VAD”

8) Line 161 that describes the source of the coverage data used in the simulation. Is there a reason why the authors did not consider using the globally available UNICEF database of national VAS coverage data by country? This data is administrative data, carefully reviewed and curated by UNICEF Data and Analytics, and publishes coverage by year, by country, by semester, and two-dose coverage –and can be found here: Vitamin A Deficiency in Children - UNICEF DATA The user can also download the full excel dataset using the link on that page. This would be the better source for actual VAS coverage in a given country in a given year, as this is also the globally accepted source for VAS coverage data included in the Global Nutrition Report (GNR). If the authors do not decide to use the UNICEF data, I recommend the reason be included in the paper.

9) As a reviewer, my expertise is in the area of VAS programs, and Vitamin A nutrition and public health and I do not consider myself qualified to comment on statistical analysis, which is why I answered Q#2 in that manner.

10) Line 326: recommend the authors provide more context to make the connection to the Edejer paper and the relevance. In addition, suggest the authors also clarify if the $85/DALY result was all of the interventions combined, or each intervention at a time etc.

11) Line 360: could the authors build this out a little bit more to describe some additional cost drivers that are relevant to VAS programs. Costs are more likely to vary due to a) the delivery mechanism/or platform used in each country to deliver the VAS to children twice/year (routine, or campaign, or what kind of campaign), and b) the strength of the health system. Some countries need to spend a lot of money to deliver VAS to every child because the health system is not already reaching them. Did the authors feel there was sufficient data on costs in the literature? If not, could the authors make a recommendation of some kind – and comment on how this would improve the quality of future analysis such as this.

Reviewer #2: Large Issues:

1. The core results emerge logically from the assumptions and modeling approach – if more children receive VAS, there will be fewer VA-preventable deaths, and if program costs are linear then the cost-effectiveness parameters look promising. However, there are reasons to question the mortality-reducing impacts of VAS in today’s world (estimates are old and new ones cannot be generated for ethical reasons), and especially to question the linearity of costs of expanding coverage. There are reasons why coverage is so low in Nigeria, for example, and these suggest increasing costs as coverage increases, and perhaps some non-cost-related thresholds that need to be considered.

2. There is a big difference between inadequate intake of VA and VAD. A thorough analysis of the impact of a VAS campaign should consider both.

3. The authors use the suggestions that (e.g.) LSFF might be a better way of delivering VA to children to justify their analyses. However, no modeling is done to capture the cost-effectiveness of alternative VA deliver platforms, so the justification lacks punch. Vosti et al. (2020) and others have done such work.

4. It is not exactly clear what the authors mean when they say VAS; what are the VAS distribution details and associated cost functions? There are many ways to manage VAS programs, and the impacts and costs vary dramatically depending on the approach chosen.

5. Most countries in sub-Saharan Africa have found campaign-based VAS to be very expensive – indeed, if the international community were not covering the costs of these campaigns (including all inputs and many of the management costs), they would not be undertaken. So, what the authors suggests as unit costs seem not to jibe with reality; indeed, the country-level estimates of program expansion costs (even after adjusting for target population size) are an order of magnitude or more smaller than estimates offered by others. See Schott et al. 2021 for an example of the structure of costs for MNP distribution – not the same intervention, but the cost structure will be similar for VAS campaigns.

6. The authors note the importance of addressing uncertainty in their modeling efforts, but uncertainty regarding the impact of VAS on VAD-attributable mortality and (especially) regarding scale-up costs seem not to have received such attention.

7. The authors contend that VAS is a highly cost-effective intervention; new literature and ongoing experience in the field suggest that this may not be the case. This is an especially contentious claim based on linear costs of scaling up VAS programs.

 

Smaller Issues:

1. It is not clear why these three countries were chosen.

2. The authors admit that there are no YLD parameters in this simulation exercise, so best to exclude them. Indeed, I would suggest presenting results in terms of deaths averted (rather than DALYs), since no comparisons are being made with other interventions with different mortality/disability structure.

3. It is not clear what ‘individual-based’ means, nor it is clear what calibrations were undertaken to ‘match’ (how well?) GBD estimates. What are SIR and SIS models? What does fixed-effect metanalysis mean, in the several places in which the term is used? Generally, the paper is not accessible to those outside the IHME and their sphere of influence; this can be fixed, but some additional effort will be required.

6. PLOS authors have the option to publish the peer review history of their article (what does this mean?). If published, this will include your full peer review and any attached files.

Reviewer #1: No

Reviewer #2: No

---

## [Author Response · Author response to Decision Letter 0]

29 Jan 2022

Response to reviewers for Vitamin A Supplementation

Reviewer 1

The paper is very well thought out, very relevant, and timely in its topic.

Thank you very much for this positive assessment of our work.

VAS is an essential child survival strategy in populations where VAD is a public health problem among children 6-59 mos of age. Despite it being very effective to reduce morbidity and mortality associated with VAD, it is an intervention that has been in place in some countries for over 20 years, and commitment to maintain such programs is waning in some countries. Cost-effectiveness data is critical to engaging policy makers, and advocating for continuation of this life-saving public health intervention especially as health systems are being stretched to the limits of their resources. For this reason we really welcome this analysis, and the rigour to which the authors brought to it.

We thank the review again!

Understanding that the authors simulated the scale up of VAS coverage from a hypothetical baseline of coverage to demonstrate that across three different settings, the effort to scale up the intervention is very cost effective.

I recommend however that this paper be revisited to consider the following:

1) The GBD 2019 significantly revised the methodology used and cited the following major changes that resulted in a very large reduction in # of DALYs due to VAD (reduced to 3.3M globally, and specifically 2.63M DALY’s in children U5):

• Vitamin A deficiency and vitamin A supplementation were modelled in ST-GPR to achieve improved time trends. 

• Vitamin A supplementation estimates are now age-sex specific since supplementation campaigns target children. 

• The age-specific stunting SEV was added as a covariate for vitamin A deficiency, alongside the three used last year: SDI, the availability of retinol activity equivalent (RAE) units in foods, and newly updated vitamin A supplementation. 

• The evidence on vitamin A deficiency as a risk factor for diarrhoea, measles, and lower respiratory infections (LRIs) was reanalysed and evaluated using MR-BRT. LRIs were removed as an outcome due to insufficient evidence, and the relative risks for diarrhoea and measles were updated. Notably, we no longer adjust relative risks for background vitamin A deficiency prevalence.

Attached to this review is the Lancet ref, as well please see this recently published discussion paper: Basis for changes in the global burden of disease estimates related to vitamin A and zinc deficiencies in the GBD 2017 and 2019 studies | Public Health Nutrition | Cambridge Core 

Could the authors consider re-running the analysis with the GBD 2019 data? If not, I strongly recommend this be acknowledged in the introduction as well as in the Discussion as a limitation, and consider what the ICER would be if the GBD 2019 was the source instead. 

We thank the reviewer for identifying this improvement to our results. We were indeed able to rerun the analysis with the GBD 2019 data and found that it substantially changed the outcomes of the simulation!

Using GBD 2019’s data, we found that the risk effects for causes of death due to VAD were significantly reduced. We also excluded LRI from our model and this contributed to the higher ICERs that we now report in our manuscript. Specifically, our model reported an ICER of $860/DALY [320, 3530] in Nigeria, $550/DALY [240, 2230] in Kenya, and $220/DALY [80, 2470] in Burkina Faso.

Because this is much larger than what others have reported previously, we revised the tables, discussion, and conclusion of our paper to reflect our new interpretation based on the model. We pointed out the changes of the latest estimates from GBD 2019 and how these inputs affected our model. We compare our results briefly with the outputs we reported earlier when we submitted a model that was calibrated with estimates from GBD 2017. For reference, below is the original set of results from GBD 2017 and GBD 2019:

Country Overall DALYs Averted Per 100,000 Person-Years Overall Additional Cost Per 100,000 Person-Years (USD / 100,000 Person-Years) Cumulative ICER (USD/DALY)

Nigeria 440 (290, 680) 17,670 (17,560, 17,730) 41 (26, 61)

Kenya 180 (120, 250) 11,340 (11,300, 11,380) 62 (46, 97)

Burkina Faso 90 (50, 140) 3,120 (3,090, 3,140) 35 (22, 59)

Table 1: Results from GBD 2017 inputs

Country Overall DALYs Averted Per 100,000 Person-Years Overall Additional Cost Per 100,000 Person-Years (USD / 100,000 Person-Years) Cumulative ICER (USD/DALY)

Nigeria 21 (5, 56) 17,880 (17,800, 17,960) 860 (320, 3530)

Kenya 21 (5, 47) 11,390 (11,350, 11,430) 550 (240, 2230)

Burkina Faso 14 (0, 37) 3,100 (3,060, 3,120) 220 (80, 2470)

Table 2: Results from GBD 2019 inputs

2) Line 67 references a paper that compared measles vaccination and therapeutic VAS in reducing mortality due to measles. While I am unable to access the full paper referenced (#10), it appears that that specific paper was considering the number of capsules given to treat a case of measles (in accordance with WHO Guidance on the therapeutic uses of VAS) – whereas the authors are comparing this to the two doses provided in PREVENTIVE programs. While preventive VAS can reduce incidence of measles by 50% (Imad et al 2017) – it is not found to reduce mortality due to measles. 

We thank the reviewer for pointing out this area for improvement in our exposition. We have updated the text and reference to make the mechanism by which measles vaccination could impact the cost-effectiveness of VAS more precise. 

3) Line 67, in addition to the above also has the following error: Large scale Food fortification does not reduce deaths associated with measles, LRI and diarrhoea – instead it is a strategy to increase the consumption of Vitamin A through the daily diet, and as such, can be considered as a means to reduce VAD in the population over time. Vitamin A supplementation on the other hand, only temporarily reduces VAD in the population, and so if a child consumes two high dose supplements, the supplement is protecting them from the effects of VAD (caused by the low intake of VA from the diet). As a result, VAS and fortification (and other dietary strategies) are complementary to each other – not alternatives. The use and interpretation of serum retinol distributions in evaluating the public health impact of vitamin A programmes | Public Health Nutrition | Cambridge Core

We thank the reviewer for identifying this point, which we have now clarified in the text. We agree that VAS and LSFF should be considered complementary approaches.

4) Line 72 – refers to VAS as a treatment – however VAS programs are a preventive public health intervention. Recommend replacing “treatment” with the term “intervention”

We thank the reviewer for pointing out this issue. We have corrected this description of VAS in the manuscript.

5) Figure 1: this would change if GBD 2019 were considered instead

As we switched to a model with GBD 2019 estimates, we have now also updated Figure 1 to reflect the model diagram without LRI as a cause of death associated with VAD.

6) Line 128 – the cohort is referred to as children 0-59 months of age – however the eligible ages for VAS is 6 months to 59 months of age.

We thank the reviewer identifying this point of confusion. Our model does track individuals between the ages of 0-59 months as more individuals with age 0 are added to the simulation based on the population birth rate. However, we only simulate the effect of VAS on children between ages 6-59 months. Our explanation of the cohort on line 128 was confusing and it has been modified to describe our model more explicitly.

7) Line 139: VAS does not treat VAD. Recommend the change to “effectiveness of VAS to reduce VAD”

We thank the reviewer for pointing out this issue. We have changed the wording to accurately reflect the relationship between VAS and VAD.

8) Line 161 that describes the source of the coverage data used in the simulation. Is there a reason why the authors did not consider using the globally available UNICEF database of national VAS coverage data by country? This data is administrative data, carefully reviewed and curated by UNICEF Data and Analytics, and publishes coverage by year, by country, by semester, and two-dose coverage –and can be found here: Vitamin A Deficiency in Children - UNICEF DATA The user can also download the full excel dataset using the link on that page. This would be the better source for actual VAS coverage in a given country in a given year, as this is also the globally accepted source for VAS coverage data included in the Global Nutrition Report (GNR). If the authors do not decide to use the UNICEF data, I recommend the reason be included in the paper. 

We thank the reviewer for pointing out this improvement for our method description. We found that the coverage in the UNICEF database for two of our countries of interest (Nigeria and Kenya) varied tremendously from year to year. If we used the coverage value from a specific year for the simulation, it was not clear to us how accurate it would be given that the measured coverage might shift largely the next year. In addition, for the third nation (Burkina Faso), the UNICEF database estimated that the coverage was 97-99 percent, which is very large and perhaps unlikely. 

For these reasons, we chose to conduct our own literature search for coverage in our countries of interest. We have included this explanation in the text.

9) As a reviewer, my expertise is in the area of VAS programs, and Vitamin A nutrition and public health and I do not consider myself qualified to comment on statistical analysis, which is why I answered Q#2 in that manner.

10) Line 326: recommend the authors provide more context to make the connection to the Edejer paper and the relevance. In addition, suggest the authors also clarify if the $85/DALY result was all of the interventions combined, or each intervention at a time etc.

We appreciate the reviewer’s advice to add more detail to this part of the discussion to justify the comparison to the Edejer paper. We have added this explanation and added some more context to the study. We also clarified that the $85/DALY result is for all the interventions combined.

11) Line 360: could the authors build this out a little bit more to describe some additional cost drivers that are relevant to VAS programs. Costs are more likely to vary due to a) the delivery mechanism/or platform used in each country to deliver the VAS to children twice/year (routine, or campaign, or what kind of campaign), and b) the strength of the health system. Some countries need to spend a lot of money to deliver VAS to every child because the health system is not already reaching them. Did the authors feel there was sufficient data on costs in the literature? If not, could the authors make a recommendation of some kind – and comment on how this would improve the quality of future analysis such as this. 

There are two changes that we make based on this recommendation. First, we clarify that the mechanism that we model in our simulation is a national campaign-based program, meant to model Child Health Days where VAS is distributed as one among many interventions. We have added this in the introduction and methods of the paper. In the discussion, we add more detail regarding the need for up-to-date and accurate cost data in the literature, and how that would improve our study.

Reviewer 2

Review of ‘Cost-effectiveness of Vitamin A Supplementation among Children in Three sub-Saharan African Countries: An Individual-based Simulation Model Using Estimates from the Global Burden of Disease 2017’

Summary Assessment: This is an interesting modeling exercise that has the potential to make useful contributions to the literature. However, substantial improvements are required to the current manuscript. 

Thank you for reviewing our paper.

Large Issues:

1. The core results emerge logically from the assumptions and modeling approach – if more children receive VAS, there will be fewer VA-preventable deaths, and if program costs are linear then the cost-effectiveness parameters look promising. However, there are reasons to question the mortality-reducing impacts of VAS in today’s world (estimates are old and new ones cannot be generated for ethical reasons), and especially to question the linearity of costs of expanding coverage. There are reasons why coverage is so low in Nigeria, for example, and these suggest increasing costs as coverage increases, and perhaps some non-cost-related thresholds that need to be considered. 

There are two parts to this, which we consider separately. Regarding the impact of VAS on mortality, we agree that ethics prohibit updating old estimates directly, and therefore we believe that by combining measured effects of VAS on VAD and the GBD-estimated effects of VAD on measles and diarrheal diseases (and maybe LRI, see response to previous reviewer about change between GBD 2017 and 2019) we have obtained the most reliable estimate of the impact of VAS on child health.

Regarding the cost, we acknowledge that our assumption of linearity is a simplification, and agree with the reviewer that the marginal costs of increasing VAS coverage will be nonlinear. Although we already acknowledged this in the Discussion section at a limitation of our model, we have revised the organization and wording to further emphasize this. We have additionally explained how a different cost estimate would affect our analysis.

2. There is a big difference between inadequate intake of VA and VAD. A thorough analysis of the impact of a VAS campaign should consider both. 

We thank the reviewer for pointing out the distinction between inadequate intake of VA and VAD. GBD focuses on VAD, although it includes data on VA intake in that estimate. Usually, strategies that involve fortification are designed to directly increase VA intake, so we did not model it in this manuscript. We have clarified this as a limitation to our model in the discussion.

3. The authors use the suggestions that (e.g.) LSFF might be a better way of delivering VA to children to justify their analyses. However, no modeling is done to capture the cost-effectiveness of alternative VA deliver platforms, so the justification lacks punch. Vosti et al. (2020) and others have done such work. 

We thank the reviewer for identifying this important issue. Although we feel that modelling other Vitamin A delivery mechanisms is important, we believe that it may be beyond the scope of this paper. We hope to address this in future work—our simulation is designed to be modular and allow an apples-to-apples comparison of LSFF and VAS approach (or even combinations!).

4. It is not exactly clear what the authors mean when they say VAS; what are the VAS distribution details and associated cost functions? There are many ways to manage VAS programs, and the impacts and costs vary dramatically depending on the approach chosen. 

We appreciate the reviewer’s help to clarify the description of our simulated distribution program and cost methods. We consider this in two parts. First, we have added in the introduction and methods of our paper that our simulation is meant to model the Child Health Days distribution program, where VAS is distributed as one among many interventions.

Additionally, our paper describes the way we use results from our simulation to calculate the overall cost of the intervention under the “Intervention Unit Cost” subsection of the methods. It illustrates what the simulation outputs (Supplemented Days), what the outputs mean, and how they are processed to determine intervention cost.

5. Most countries in sub-Saharan Africa have found campaign-based VAS to be very expensive – indeed, if the international community were not covering the costs of these campaigns (including all inputs and many of the management costs), they would not be undertaken. So, what the authors suggests as unit costs seem not to jibe with reality; indeed, the country-level estimates of program expansion costs (even after adjusting for target population size) are an order of magnitude or more smaller than estimates offered by others. See Schott et al. 2021 for an example of the structure of costs for MNP distribution – not the same intervention, but the cost structure will be similar for VAS campaigns. 

We thank the reviewer for their suggestion to check the unit cost for our model to be consistent with other estimates. Because the unit cost directly affects our ICER calculations, it is important to use a reliable value. Based on the literature for VAS costs, we believe that our unit cost is appropriate. Below we include a table describing estimates for VAS unit costs for campaign-based programs in sub-Saharan African countries.

Paper Cost Notes

Neidecker-Gonzales et al. 2007 0.51 USD (Ghana)

0.61 USD (Zambia) Other countries are also considered in this literature review. Unit costs vary tremendously from country to country, so we decided only to use values from sub-Saharan African countries.

Kagin et al. 2015 0.51 USD (Cameroon) The costs analysis in this study found that the wages, training, and communication took up most of the cost. The cost per capsule was 0.03 USD.

Horton et al. 2018 0.62 USD (Senegal) This paper reported the cost as 728.5 FCFA per child. Here, we used the conversion provided in the paper of 1 USD = 584 FCFA and divide by 2 for a cost per dose.

This paper also finds the largest share of the cost comes from health workers’ wages.

Although we do not have data specifically for our regions of interest, we believe these estimates are appropriate. There are examples for cost varying from country to country based on budget and wages for health workers. Neidecker-Gonzales et al. 2007 hypothesize an interesting relationship between unit cost and GDP in their paper:

Because the countries in our analysis are towards the left end of the graph, we believe that this reinforces our cost estimate.

We also found that VAS tends to be much cheaper than MNP. MNP consists of 15 micronutrients including iron, zinc, folic acid, Vitamins A, C, D, B1, B6, etc., so the cost of the supplements would be larger. According to Schott et al. 2021, the cost of a sachet is 0.25-0.59 USD, and the cost of a packet is 7.40-17.83 USD per child. In addition, Kagin et al. 2015 compares the costs of VAS and MNP; they find that while it costs 0.06 USD for two VA capsules themselves, the cost of one year of MNP packets is 3.60 USD. 

There are other reasons why estimates for MNP may be higher than VAS. We considered VAS in the context of Child Health Days (CHD), which involve the distribution of many different interventions in addition to VAS—this perhaps reduces the average wage and management cost reported for VAS. Another possibility may be that Schott et al. 2021 considers opportunity costs (only one of the three papers for VAS do), which is estimated to be 17-20% of the cost they report.

In going back and looking for more estimates on unit cost data, we found that the WHO One Health Tool (OHT) provided an estimate of 2.45-3.17 USD. As we believed that the assumptions of this estimate differ substantially from our scenario for CHD, we did not change our unit cost estimate, but we added some reflection on it in the limitations section.

Getting the unit cost right is very important for this analysis as it directly affects the ICER results of the simulation. Although we believe that our current cost estimate agrees with the values in the evidence base at this time, we appreciate the reviewer’s inquiry about our unit cost, and we have added more detail in our discussion to explain the limitations of our cost model. We have highlighted the variability in the unit cost from the WHO OHT as well as different subnational regions in the country. We also added an explanation of how changing the unit cost would proportionally change the ICER estimate.

6. The authors note the importance of addressing uncertainty in their modeling efforts, but uncertainty regarding the impact of VAS on VAD-attributable mortality and (especially) regarding scale-up costs seem not to have received such attention. 

We address this issue in two parts. For the impact of VAS on the pipeline, we model the effect of VAS on VAD, the effect of VAD on causes of disease, and the effect of causes on DALYs. The effect of VAS on VAD is determined by a meta-analysis that includes uncertainty. The final two parts of the pipeline are taken from GBD estimates (which include uncertainty estimates). 

For the scale-up costs, we did not include uncertainty as part of the scale-up schedule because we modelled the interventions as step functions. To keep it simple, we assumed certainty in the scale-up schedule. For the uncertainty of the unit cost estimate, the paper that provided this cost estimate did not include uncertainty, so we did not add an uncertainty interval either.

For the outcomes of the simulation, we have added an explanation regarding the uncertainty calculation for the outcomes (including DALYs averted and scale-up cost) in our model under “Analysis of Simulation Results” in the methods section.

7. The authors contend that VAS is a highly cost-effective intervention; new literature and ongoing experience in the field suggest that this may not be the case. This is an especially contentious claim based on linear costs of scaling up VAS programs. 

We thank the reviewer for pointing out this issue. After changing our modelling to be in line with GBD 2019 estimates (please see Reviewer 1’s first suggestion), our simulation outcomes suggest that the intervention is actually less cost effective than what was reported in previous studies. We have modified our discussion and conclusion to analyze these new results.

 

Smaller Issues:

1. It is not clear why these three countries were chosen.

We thank the reviewer for pointing out this issue. We chose these three countries because they had different coverage levels for VAS, so the outcomes of their scale-ups would serve as an interesting point of comparison across countries. We have now stated this purpose in the introduction to make the rationale behind the choice of countries clearer.

2. The authors admit that there are no YLD parameters in this simulation exercise, so best to exclude them. Indeed, I would suggest presenting results in terms of deaths averted (rather than DALYs), since no comparisons are being made with other interventions with different mortality/disability structure. 

We appreciate the reviewer’s suggestion to present the ICER value to be in cost per death averted. Although we do not incorporate YLDs for xerophthalmia, we include YLDs for diarrheal diseases (as well as LRI in the GBD 2017 model). These YLDs are included in our DALYs estimate. Although we agree that DALYs are often unclear, we prefer to report ICER in dollars per DALY in our analysis as it is the conventional unit used in the literature.

3. It is not clear what ‘individual-based’ means, nor it is clear what calibrations were undertaken to ‘match’ (how well?) GBD estimates. What are SIR and SIS models? What does fixed-effect metanalysis mean, in the several places in which the term is used? Generally, the paper is not accessible to those outside the IHME and their sphere of influence; this can be fixed, but some additional effort will be required. 

We thank the reviewer for identifying that clarifying terminology would make this paper accessible to a wider audience. We have added explanations for these terms in our revisions.

---

## [Decision Letter · Decision Letter 1]

11 Mar 2022

PONE-D-21-34978R1Cost-effectiveness of Vitamin A Supplementation among children in three sub-Saharan African countries: an individual-based simulation model using estimates from Global Burden of Disease 2019PLOS ONE

Dear Dr. Kannan,

Thank you for submitting your manuscript to PLOS ONE. After careful consideration, we feel that it has merit but does not fully meet PLOS ONE’s publication criteria as it currently stands. Therefore, we invite you to submit a revised version of the manuscript that addresses the points raised during the review process.

Reviewer 1 has noted two outstanding issues regarding the data. Please address these issues in your revisions.

We look forward to receiving your revised manuscript.

Kind regards,

Susan Horton

Academic Editor

PLOS ONE

Journal Requirements:

Reviewers' comments:

Reviewer's Responses to Questions

**Comments to the Author**

1. If the authors have adequately addressed your comments raised in a previous round of review and you feel that this manuscript is now acceptable for publication, you may indicate that here to bypass the “Comments to the Author” section, enter your conflict of interest statement in the “Confidential to Editor” section, and submit your "Accept" recommendation.

Reviewer #1: All comments have been addressed

Reviewer #2: All comments have been addressed

2. Is the manuscript technically sound, and do the data support the conclusions?

Reviewer #1: Partly

Reviewer #2: Yes

3. Has the statistical analysis been performed appropriately and rigorously? 

Reviewer #1: I Don't Know

Reviewer #2: Yes

4. Have the authors made all data underlying the findings in their manuscript fully available?

Reviewer #1: Yes

Reviewer #2: Yes

5. Is the manuscript presented in an intelligible fashion and written in standard English?

Reviewer #1: Yes

Reviewer #2: Yes

6. Review Comments to the Author

Reviewer #1: One outstanding question for me: The VAD prevalence data indicated in Table 1 seems off by quite a bit - for instance WHO estimates for VAD among PRESAC children in Nigeria is in the range of 15-25% -- whereas Table 1 states it as 1.2%. What is the data source for the VAD prevalence used in the analysis? GBD 2019 also indicates the analysis no longer corrects for background VAD - whereas the explanation of the difference in results is partially due to the differences in VAD prevalence. Please review before proceeding.

Reviewer #2: The updated manuscript is much improved; indeed, it is much clearer and the conclusions have been completely reversed! There are still some additional work that could be done (e.g., additional sensitivity analysis regarding some key model parameters and assumptions), but to my mind, the authors have met the criteria for publication. Congrats to all!

7. PLOS authors have the option to publish the peer review history of their article (what does this mean?). If published, this will include your full peer review and any attached files.

Reviewer #1: No

Reviewer #2: No

---

## [Author Response · Author response to Decision Letter 1]

20 Mar 2022

Response to reviewers for Vitamin A Supplementation

Reviewer 1

One outstanding question for me: The VAD prevalence data indicated in Table 1 seems off by quite a bit - for instance WHO estimates for VAD among PRESAC children in Nigeria is in the range of 15-25% -- whereas Table 1 states it as 1.2%. What is the data source for the VAD prevalence used in the analysis? GBD 2019 also indicates the analysis no longer corrects for background VAD - whereas the explanation of the difference in results is partially due to the differences in VAD prevalence. Please review before proceeding.

We thank the reviewer for pointing out this issue in Table 1. We reviewed this issue and found that indeed the prevalence for VAD in Table 1 deviated from previous estimates. 

These values were generated from the GHDx GBD Results tool. We found that the values for VAD prevalence were not updated properly in GHDx for GBD 2019 due to a last-minute resubmission. We updated Table 1 to accurately reflect the inputs for VAD prevalence from GBD 2019. We also added prevalence values for Measles and Diarrheal Diseases that were inputs in the simulation.

Fortunately, the values for VAD risk exposure that we used for the simulation were updated already to include the correct GBD estimates, so this change does not affect the outputs of the simulation.

Reviewer 2

The updated manuscript is much improved; indeed, it is much clearer and the conclusions have been completely reversed! There are still some additional work that could be done (e.g., additional sensitivity analysis regarding some key model parameters and assumptions), but to my mind, the authors have met the criteria for publication. Congrats to all!

Thank you very much for this positive assessment of our work.

---

## [Editor Report · Decision Letter 2]

22 Mar 2022

Cost-effectiveness of Vitamin A Supplementation among children in three sub-Saharan African countries: an individual-based simulation model using estimates from Global Burden of Disease 2019

PONE-D-21-34978R2

Dear Dr. Kannan,

We’re pleased to inform you that your manuscript has been judged scientifically suitable for publication and will be formally accepted for publication once it meets all outstanding technical requirements.

Kind regards,

Susan Horton

Academic Editor

PLOS ONE
---

## [Editor Report · Acceptance letter]

29 Mar 2022

PONE-D-21-34978R2 

Cost-effectiveness of Vitamin A Supplementation among children in three sub-Saharan African countries: an individual-based simulation model using estimates from Global Burden of Disease 2019 

Dear Dr. Kannan:

I'm pleased to inform you that your manuscript has been deemed suitable for publication in PLOS ONE. Congratulations! Your manuscript is now with our production department. 

Kind regards, 

on behalf of

Dr. Susan Horton 

Academic Editor

PLOS ONE